# Acute Myeloid Leukemia Mutations: Therapeutic Implications

**DOI:** 10.3390/ijms20112721

**Published:** 2019-06-03

**Authors:** Cristina Papayannidis, Chiara Sartor, Giovanni Marconi, Maria Chiara Fontana, Jacopo Nanni, Gianluca Cristiano, Sarah Parisi, Stefania Paolini, Antonio Curti

**Affiliations:** Istituto di Ematologia e Oncologia Medica “L. e A. Seràgnoli”, S.Orsola-Malpighi Hospital, 40138 Bologna, Italy; chiara.sartor@studio.unibo.it (C.S.); giovanni.marconi@studio.unibo.it (G.M.); mariachiara.fontana4@unibo.it (M.C.F.); jacopo.nanni2@studio.unibo.it (J.N.); gianluca.cristiano2@studio.unibo.it (G.C.); sarah.parisi2@unibo.it (S.P.); stefania.paolini@unibo.it (S.P.); antonio.curti2@unibo.it (A.C.)

**Keywords:** acute myeloid leukemia, mutations, FLT3, IDH1-2, resistance

## Abstract

Acute Myeloid Leukemia (AML) is an extremely heterogeneous group of hematological neoplasms, for which allogeneic stem cell transplantation (HSCT) still represents the only potentially curative option in the majority of cases. However, elderly age and clinically severe comorbidities may often exclude a wide amount of patients from this therapeutic approach, underlying the urgent need for alternative strategies. Thanks to the introduction of advanced high-throughput techniques, light is being shed on the pathogenesis of AML, identifying molecular recurrent mutations as responsible for the onset, as well as progression, of disease. As a consequence, and in parallel, many new compounds, including targeted therapies (FMS-like tyrosine kinase 3 (FLT3) and Isocitrate dehydrogenase 1-2 (IDH1-2) inhibitors), have found a wide room of application in this setting, and are now available in daily practice, or in late phases of clinical development. Moreover, several further innovative molecules are currently under investigation, and promising results for many of them have already been reported. In this review, we will present an update on the most relevant molecular alterations of AML, focusing on the most frequent genomic mutations of the disease, for which compounds have been approved or are still currently under investigation.

## 1. Introduction

Acute Myeloid Leukemia (AML) is a group of hematological malignancies characterized by the rapid and uncontrolled growth of immature white blood cells, which accumulate in the bone marrow and alter normal hematopoiesis [1]. This disease accounts for approximately 30% of leukemias in the adult setting, involving mainly elderly patients (median age at diagnosis 65 years), and reaching the peak at around 80 years of age [2]. Despite relevant advances having been made in the understanding of AML pathogenesis, standard chemotherapy followed by allogeneic stem cell transplantation (HSCT) still represents the only potentially curative approach in the majority of cases, with five-year survival being rare, ranging from 50 to 70% [3]. However, elderly age and related clinical comorbidities may often exclude a wide number of patients from the use of high-dose cytotoxic chemotherapy, leading to a long-term survival rate in this setting not exceeding 10%. As a consequence, in this context, the identification of novel molecular alterations, potentially suitable for targeted therapies, is strongly required. To this aim, next-generation sequencing (NGS) approaches have dissected AML pathogenesis, and showed a quite remarkable genomic complexity and heterogeneity, thus leading to the identification of multiple signaling pathways involved in disease onset and progression [4]. From a therapeutic perspective, after a long absence of innovative compounds in the setting of AML, new drugs have recently been approved by FDA. Among these, midostaurin (an oral FMS-like tyrosine kinase 3 (FLT3) inhibitor) and enasidenib (an oral Isocitrate dehydrogenase 2 (IDH-2) inhibitor) are directed on two mutations detected in a significant amount of AML patients.

In this review, we will report on the most relevant and frequent molecular mutations detected in AML, and we will summarize the results obtained with the drugs recently approved, or currently still in clinical development, in this setting.

## 2. Mutational Scenario in AML

AML originates from a hematopoietic stem or progenitor cell (HSPC) that gains genomic and molecular alterations. As a consequence, HPSC acquires stem-cell like properties of unrestricted self-renewal, becoming capable of maintaining the malignant neoplastic clone [5].

Thanks to improvements in current technologies, represented by chromosome banding, fluorescence in situ hybridization/chromosomal painting, array comparative genomic hybridization, genomic breakpoint cloning, Sanger Sequencing of candidate genes, single nucleotide polymorphism profiling, whole genome sequencing (WGS), whole exome sequencing (WES), and RNA sequencing, the genetic basis of AML has been deeply dissected, and new knowledge has emerged in the last five years [6].

Overall, as reported by an in-depth analysis of a wide cohort of patients, the AML genome is characterized by a lower number of mutations, if compared to the majority of other tumors, showing an average of 13 coding mutations (single nucleotide variants and insertion/deletions) per person. These mutations usually involve deregulated pathways, such as DNA methylation-associated genes, spliceosome-complex genes, cohesion-complex genes, chromatin-modifying genes, and signaling genes. About 96% of AML patients present at least one driver mutation in one of these genes. [7,8].

Such a deep genomic complexity may at least partially explain the extreme heterogeneity in clinical onset, morphological phenotype, and consequent outcome of AML patients.

In terms of classification, the most frequent AML mutations may be summarized in Table 1. The prognostic impact of many mutations is context-dependent, with the effect related to the concomitant presence or absence of other molecular abnormalities [9]. As widely known and reported, the NPM1 mutation is associated with a “favorable” prognosis only in the absence of an FLT3-ITD clone (or FLT3-ITD with a low allelic ratio, even if, in this last subset, the prognostic impact, and the most appropriate therapeutic approach, have not yet been unanimously defined) [10]. On the other hand, TP53 mutations strongly correlate with an adverse outcome, and mainly occur in secondary or therapy-related AML, mostly characterized by complex cytogenetics [8]. The most relevant genomic mutations in AML are reported in this review, highlighting the potential therapeutic approaches, approved or in clinical development, to these abnormalities.

## 3. FLT3 Mutations

The FLT3 receptor has been reported to be mutated in about 30% of AML patients [11], and two kinds of mutations may be identified: in-frame duplications within the juxtamembrane region (FLT3-ITD), which can be observed in about 25% of AML, and point mutations in the tyrosine kinase domain (FLT3-TKD), detected in 7% of the cases. Both types of mutations induce a constitutive activation of the receptor, even if their prognostic impact seems to be different. In detail, rat sarcoma (RAS) and phosphoinositide 3-kinase (PI3K) signaling pathways are activated by the mutant forms, leading, mainly in FLT3/ITD mutant, to a reduced expression of SH2 domain-containing protein tyrosine phosphatase 1 (SHP-1), and inducing a block of cell differentiation and cell death by apoptosis [12,13,14,15,16]. In terms of prognosis, the FLT3/ITD mutations (particularly those with a high allelic burden) are associated with an adverse outcome, in terms of complete response rates and long-term survival. On the contrary, the impact on prognosis of FLT3/TKD mutations is actually still debated and controversial [17,18,19], being frequently reported to be associated with better survival. The biological reason for such an opposite outcome may be found in the concomitant presence of additional cooperating mutations, which may have an impact on the disease aggressiveness and chemo-resistance.

### 3.1. FLT3 Inhibitors

In order to translate this biological knowledge into clinical practice, FLT3 inhibitors have been designed and developed, to be tested first in the relapsed/refractory setting, and, more recently, in combination with standard chemotherapy in the front-line setting. According to their specificity for FLT3, we can distinguish first- and second-generation agents.

#### 3.1.1. First-Generation FLT3 Inhibitors: Midostaurin

The first inhibitors are multi-targeted kinase inhibitors (midostaurin, sorafenib, lestaurtinib), which demonstrated, in in vitro settings, a good inhibition of mutant FLT3. Among these, midostaurin is a pan-kinase inhibitor, which demonstrated in vivo activity against, among others, KIT and FLT3, in their wild type and mutated forms [20]. The drug activity in AML was explored first in early-phase clinical studies, whose promising results [21] led to the design of the CALGB10603/RATIFY phase 3 trial. This was a multicenter, double-blind placebo-controlled study, which enrolled 717 adult patients (18–59 years of age) affected by newly diagnosed FLT3-mutated AML [22], randomized to receive either a standard program of chemotherapy based on induction (daunorubicin + cytarabine) and consolidation (high-dose cytarabine) plus a placebo (*n* = 357) or midostaurin (*n* = 360), at a dosage of 50 mg twice daily on days 8 through 21; after the end of consolidation therapy, patients who were still in remission went on to receive midostaurin or a placebo for 12 months. In terms of result, patients enrolled in the midostaurin arm group reached a prolonged event-free survival (EFS, 8.2 months versus 3.0 months; *p* = 0.002), disease-free survival (DFS, 26.7 months versus 15.5 months; *p* = 0.01), and OS (74.7 months versus 25.6 months; *p* = 0.009). Furthermore, a 22% reduced risk of death (HR = 0.78, *p* = 0.009) and a 21.6% lower risk of relapse (HR = 0.78, *p* = 0.002) in the experimental arm than in the placebo group were observed. In multivariate analysis, a benefit in terms of EFS and OS was seen in the midostaurin arm regardless of the FLT3 mutation type (TKD or ITD) and from the allelic ratio (high or low). As for drug response, despite the CR rate (CR reported within 60 days of protocol therapy initiation) was only slightly higher in the midostaurin arm than in the placebo one (58.9% and 53.5%, respectively, *p* = 0.15), taking into account all the CRs obtained during treatment and within 30 days of treatment discontinuation, the CR rate was significantly higher in the experimental group of patients (68% versus 61%; *p* = 0.04). Furthermore, a higher rate of patients in the midostaurin arm was able to proceed to allogenic HSCT in first remission (28.1% versus 22.7% respectively) (*p* = 0.10). As far as safety was concerned, in the midostaurin arm grade ≥ 3 anemia and skin rash were more common if compared with the placebo arm. The incidence of all the other adverse events was similar between the two groups. The results of this trial led FDA and EMA to approve midostaurin for the treatment of newly-diagnosed, FLT3-mutated AML patients, in combination with standard chemotherapy, thanks to the survival improvement reached, for the first time, thanks to the introduction of a targeted agent to a conventional chemotherapy-based approach. The clinical development of midostaurin is still ongoing, and other studies have been designed, exploring the combination with decitabine in elderly AML patients (NCT01846624, recently closed to the enrollment), or the potential role of the drug in FLT3-negative AML patients (NCT03512197) exploiting its wide spectrum of action, directed to many further molecular targets, besides FLT3.

#### 3.1.2. Second-Generation FLT3 Inhibitors: Quizartinib, Crenolanib, and Gilteritinib

Second-generation FLT3 inhibitors include quizartinib, crenolanib, and gilteritinib, and show a more selective inhibitory activity, as well as a higher potency, if compared to first-generation compounds. The preliminary phase 1 studies on quizartinib led to the assessment of the maximum tolerated dose of the compound, which was fixed at 200 mg/day, demonstrating a high efficacy in terms of response rates in the relapsed/refractory patients population [23]. Therefore, several phase II studies have been conducted in the same setting [24,25], confirming the efficacy and good tolerance of a single-agent quizartinib appriach, as a promising tool to reach a better outcome in patients with such a dismal prognosis. Based on these data, a phase III, open-label, randomized clinical trial, Quantum-R (NCT02039726), was designed, exploring the administration of quizartinib versus salvage chemotherapy (SC) in patients with relapsed or refractory, FLT3/ITD-positive AML, after first-line treatment with or without HSCT. The final data have recently been presented, demonstrating that the drug led to a significant reduction in the risk of death by 24% compared to SC. The efficacy of the drug was also evaluated in clinical trials exploring the combination with standard chemotherapy, showing promising results [26]; recently, a phase III trial with quizartinib versus a placebo as first line approach in combination with “3 + 7” induction and consolidation therapy has been designed, and it’s currently ongoing in many centers worldwide (NCT02668653). Based on the published data, in August 2018, FDA granted breakthrough therapy designation to quizartinib for relapsed/refractory FLT3-ITD AML.

Crenolanib is a pan TKI inhibitor, with demonstrated activity not only on FLT3, but also, among others, on platelet-derived growth factor receptors (PDGFR) [27]. Both preclinical and clinical data showed that the compound is highly active on D835-FLT3-mutated patients, underlying a potentially relevant role of the drug after other FLT3 inhibitor failures. In detail, in a phase II study enrolling relapsed/refractory FLT3-mutated AML patients, it was shown that the drug (dosage of 200 mg, three times/day continuously in 28-day cycles) was able to induce a CR with an incomplete count recovery (CRi) in 23% of FLT3/tyrosine kinase inhibitor (TKI) naïve patients, and in 5% of patients previously treated with other FLT3 inhibitors [28]. As a consequence, the drug activity was also investigated in combination with standard chemotherapy in FLT3-mutated newly-diagnosed AML patients (crenolanib at the dosage of 100 mg, three times/day; chemotherapy according to “3 + 7” schedule), leading to a CR rate with a full count recovery of 96%. Interesting biological data on the drug have recently been reported, showing that adding crenolanib can overcome the poor prognostic implication of adverse mutations co-occurring with mutated FLT3. Therefore, these data represent the rationale to combine crenolanib with chemotherapy to improve the overall outcome of FLT3-mutated AML with diverse mutational profiles [29]. More recently, a direct comparison trial between crenolanib and midostaurin in combination with standard chemotherapy started to be explored within a clinical trial enrolling newly-diagnosed FLT3-mutant AML (NCT03258931).

Gilteritinib, also known as ASP-2215, is a pyrazinecarboxamide derivative with dual selective activity on FLT3 mutations and AXL, a member of the TAM (TYRO 3, AXL, and MER) receptor tyrosine kinase subfamily, which is able to modulate FLT3 activity [30]. The first and significant data on the drug activity came from the CHRYSALIS trial, an open-label phase I/II study that assessed the safety and tolerability, pharmacokinetic and pharmacodynamic profiles, and antileukemic activity of gilteritinib in a large cohort of relapsed/refractory AML patients (*n* = 252) [31]. The maximum tolerated dose was fixed at 300 mg, and clinical activity was observed at a dose of ≥80 mg once daily, in patients with FLT3-mutated R/R AML. As for toxicity, the most common grade 3–4 adverse events, irrespective of relation to treatment, were febrile neutropenia (97 (39%) of 252), anemia (61 (24%)), thrombocytopenia (33 (13%)), sepsis (28 (11%)), and pneumonia (27 (11%)). Commonly reported treatment-related adverse events were diarrhea (92 (37%) of 252), anemia (86 (34%)), fatigue (83 (33%)), elevated aspartate aminotransferase (65 (26%)), and increased alanine aminotransferase (47 (19%)). Serious adverse events occurring in 5% or more of patients were febrile neutropenia (98 (39%) of 252; five related to treatment), progressive disease (43 (17%)), sepsis (36 (14%); two related to treatment), pneumonia (27 (11%)), acute renal failure (25 (10%); five related to treatment), pyrexia (21 (8%); three related to treatment), bacteraemia (14 (6%); one related to treatment), and respiratory failure (14 (6%)). An exposure-related increase in the inhibition of FLT3 phosphorylation was noted with increasing concentrations of gilteritinib in plasma. As for efficacy, the overall response rate was 100 (40%), with 19 (8%) achieving complete remission, ten (4%) achieving complete remission with incomplete platelet recovery, 46 (18%) achieving complete remission with incomplete haematological recovery, and 25 (10%) achieving partial remission. Starting from these results, many phase III studies have been designed, in order to compare gilteritinib to standard salvage chemotherapies in AML patients with FLT3 mutations in a relapsed/refractory setting (NCT02421939, NCT03182244). In particular, the positive and encouraging results of the interim analysis of the ADMIRAL trial led the drug to gain FDA approval for R/R FLT3-positive AML patients in November 2018. Moving to another setting of patients, a phase I study is now recruiting newly-diagnosed AML patients primarily to define the safety and tolerability profile of gilteritinib administered in combination with induction and consolidation chemotherapy (NCT02236013).

### 3.2. Mechanisms of Resistance to FLT3 Inhibitors

Starting from the Philadelphia positive leukemia model, in which mechanisms of resistance to TKIs have been widely investigated and described [32], due to their clinical implications in patients’ management, the same topic has also been addressed in AML, in order to investigate the variable sensitivity of FLT3 inhibitors between different activating point mutations in the kinase domain of FLT3. Many models have been described, showing various potential mechanisms responsible for resistance to FLT3 inhibitors, including the desensitization of drug targets by FLT3 gene amplification or protein overexpression; decreased drug binding affinity by mutation, deletion, or insertion in TKD; increased drug efflux by p-gp; and activation of survival and proliferative pathways (molecules) such as the RAS pathway, STAT pathway, anti-apoptotic Bcl-xL, Survivin, and DNA repair molecule RAD51 [33].

Thanks to this biological knowledge, and in order to overcome these mechanisms of resistance, the use of combined approaches of FLT3 inhibitors with chemotherapy or other small molecule inhibitors targeting mTOR, HDAC, HSP90, STAT3, Bcl2, PIM family, and IAPs (Survivin and Smac) may be a successful strategy, at least in terms of biological rationale. In the near future, hopefully, the application of deep molecular approaches may help physicians in the identification of the personal oncogenic signature of every single patient, in order to design, from the beginning of every therapeutic path, the most appropriate treatment strategy.

## 4. IDH1-IDH2 Mutations

IDH1 and IDH2 are two enzymes located in the cytoplasm/peroxysomes and mitochondria, respectively. They play a role in the catalyzation of the oxidative decarboxylation of isocitrate to α-ketoglutarate. In several malignancies, from solid to hematological tumors, including AML, point mutations in these proteins were described. In particular, IDH1 and IDH2 mutations have been reported in about 20% of all AML cases, occurring in 7–14% and 8–19% of patients, respectively [34]. The metabolic consequences of these mutations in AML, which have been observed and described in in vitro and in vivo studies, showed that these alterations convert α-ketoglutarate to D-2-hydroxyglutarate (D-2HG) [35]. As a result, D-2HG interferes with cellular metabolism and epigenetic regulation, leading to an arrest of cell differentiation and the promotion of oncogenesis. Many innovative compounds have been developed to target leukemic cells carrying IDH1/2 mutations (AG-120, AG-221, BAY1436032, IDH305, AG-881, FT-2102) [36], the majority of which are still in clinical development.

### 4.1. IDH1 Inhibitors: Ivosidenib

The specific IDH1 inhibitor, ivosidenib (AG-120), was tested, as a single agent, in a phase 1 clinical trial for patients with IDH mutation-positive advanced hematologic malignancies, including refractory/relapsed AML [37]. The maximum tolerated dose was not defined, and at all the drug doses tested, a reduction of the D-2HG plasma level was observed. In terms of safety, the most common grade ≥ 3 AEs (≥15%) were febrile neutropenia, anemia, leukocytosis, and pneumonia. In terms of efficacy, the ORR was 38.5% (30/78 patients), with a CR rate of 17.9%. Notably, a clearance of mutant IDH1 was detected, by NGS VAF analysis, in 27.3% of patients who obtained a CR [38]. In a recent phase I study, CR, CR + CRi, and ORR were observed, respectively, in 21.6%, 30.4%, and 41.6% of patients with IDH1-mutated AML receiving ivosidenib 500 mg monotherapy, with a median duration of 8.2 months, 9.3 months, and 6.5 months [39]. These results led to FDA approval in July 2018, in adult patients with relapsed or refractory acute myeloid leukemia with a susceptible IDH1 mutation.

### 4.2. IDH2 Inhibitors: Enasidenib

Enasidenib (AG-221) is an oral, selective inhibitor of mutant IDH2, that was first studied in a human phase 1/2 clinical trial, exploring the safety and tolerability of the drug in R/R AML patients. In this study, 41% of 239 enrolled patients developed grade 3–4 treatment-emergent adverse events, represented by hyperbilirubinemia (12%), differentiation syndrome (6%), and thrombocytopenia (6%). In terms of clinical efficacy, which was assessed in an expansion phase (dose of 100 mg once daily), the ORR was 40.3%, and the CR rate accounted for 19.3%. The median OS was 9.3 months in all of the relapsed/refractory patients, and 19.7 months in those patients with relapsed/refractory disease who reached a CR [40]. Starting from these promising data, enasidenib was recently approved by the FDA in advanced mutant IDH2 AML. Moreover, a multicenter, open-label, randomized, phase III study IDHENTIFY is currently ongoing worldwide, and recruiting elderly subjects (≥60 years) with IDH2-mutated advanced AML, to compare enasidenib treatment to conventional care regimens (NCT02577406). Furthermore, the combination of both AG-120 and AG-221 with standard induction and consolidation therapy is currently being investigated in a phase I study in patients with IDH-mutant AML (NCT02632708), in order to explore both the safety and efficacy of a more targeted and innovative first line approach to AML.

### 4.3. Mechanisms of Resistance to IDH1-2 Inhibitors

Two recent papers have elucidated the mechanism of action responsible for the loss of response to IDH inhibitors [41]. The authors observed that, in a minority of cases, it can be due to a failure to suppress 2-HG production, because of the acquisition of a second-site mutation in the other IDH2 allele, resulting in the loss of activity of the inhibitor on the target, or to the acquisition of a mutation in IDH1, causing the production of 2-HG by IDH1 dimers that are not the target of the drug. More often, however, acquired clinical resistance is explained by the genetic or epigenetic evolution or selection of leukemic clones bearing mutations in other genes. Furthermore, the co-mutational landscape prior to treatment seems to have an impact on the response to therapy that we may observe, as well as on the pattern of clonal hematopoietic reconstitution that does occur during the differentiation response. Therefore, the concept of AML heterogeneity plays a key role, as well as its adaptability to therapeutic selection pressure. Hopefully, the development of precise approaches will help physicians to design specific treatments for every patient, in order to block mechanisms of escape from drug activity.

## 5. Conclusions

AML is a complex disease with a complex genetic landscape. Our knowledge on this field is rapidly expanding, and thanks to innovative molecular approaches, we are able to deeply characterize every disease, in every single patient. Although the median number of mutations identified in AML is lower than in many other tumors, these kinds of alterations play a key role not only in terms of pathogenesis, but also in the practical approach to cure our patients.

How can we translate these biological acquisitions into clinical practice? May they have an impact on our medical decisions? Until a couple of years ago, when experimental and innovative drugs were available only within clinical trials, at few sites worldwide, a complete and accurate molecular analysis might have been relevant only in terms of prognostic impact, without practical implications for patients’ treatment. Over the last years, notably, data from the already mentioned trials, have shown positive results, which led new molecules to be approved, as first line in combination with chemotherapy, or as single agents in relapsed/refractory populations. This being said, the search for FLT3, IDH1, and IDH2 mutations is now mandatory for physicians who cure AML, due to the implications that these alterations may have on clinical patients’ management, and to the availability of these drugs not only in the setting they were approved for, but also in more recently designed clinical trials (Table 2).

The deepest high-throughput molecular studies are strongly encouraged within research programs, in order to increase our understanding of AML and to identify further innovative compounds to be directed against other mutations.

Therefore, relevant progress has been made so far, but the path towards an AML cure still requires many efforts, based on a double approach: molecular characterization of the disease on one side, and development of new agents on the other side. The future of AML therapy will be more and more based on combinations of new drugs with standard chemotherapy in young patients, and hypomethylating agents and Bcl-2 inhibitors in the elderly. By these approaches, a decreased toxicity and a better tolerability of treatments are expected, as well as higher responses and improved results in terms of quality of life.

## Figures and Tables

**Table 1 ijms-20-02721-t001:** Mutational scenario in AML.

Mutation	Frequency in CN-AML	Targeted Agents Available?	Prognostic Impact	Drugs
NPM1	30–45%	No	Favorable	NA
DNMT3A	34%	No	Not defined	NA
FLT3-ITD	28–34%	Yes	Unfavorable in high ratio	Sorafenib, Quizartinib, Gilteritinib, Midostaurin
FLT3-TKD	11–14%	Yes	Neutral	Midostaurin, Gilteritinib, Quizartinib
IDH1/2	15–30%	Yes	Favorable	Ivosidenib, Enasidenib
TET2	10%	No	Not defined	NA
ASXL1	5–16%	No	Unfavorable	NA
CEBPA	10–18%	No	Favorable	NA
RAS	25% NRAS, 15% KRAS	Yes	Neutral	Cobimetinib
KIT	20–30% of CBF-AML	Yes	Unfavorable	Dasatinib, Imatinib
KMT2A-PTD	5–10%	No	Unfavorable	NA
RUNX1	5–13%	No	Unfavorable	NA
TP53	5–20%	Yes (wild type forms)	Unfavorable	Idasanutlin

NA—not available.

**Table 2 ijms-20-02721-t002:** FLT3 and IDH1-2 inhibitors clinical trials.

NCT	DRUG	Phase	Setting	Results	Enrollment
01846624	Midostaurin + decitabine	II	Newly diagnosed, elderly FLT3 ITD/TKD AML	NA	Closed
03512197	Midostaurin + chemotherapy	III	Newly diagnosed, >18 years, FLT3 negative AML	NA	Ongoing
02039726 [42]	Quizartinib vs. Salvage chemoterapy	III	Relapsed/refractory, >18 years, FLT3 ITD AML	Median OS: 6.2 months vs. 4.7 months; Estimated survival probability at 1 year: 27% vs. 20%	Active, not recruiting
02668653	Chemotherapy + quizartinib/placebo	III, randomized, double blind, placebo-control	Newly diagnosed, >18 years, FLT3 ITD AML	NA	Ongoing
03258931	Crenolanib vs. Midostaurin following induction and consolidation chemotherapy	III	Newly diagnosed, >18 years, FLT3 ITD/TKD AML	NA	Ongoing
02421939 [43]	Gilteritinib vs. salvage chemotherapy	III	Relapsed/refractory, >18 years, FLT3 ITD/TKD AML	Median OS: 9.3 months vs. 5.6 months; 1 year survival rate: 37% vs. 17%	Active, not recruiting
03182244	Gilteritinib vs. salvage chemotherapy	III	Relapsed/refractory, >18 years, FLT3 ITD/TKD AML	NA	Ongoing
02236013 [44]	Gilteritinib + chemotherapy	I	Newly diagnosed FLT3 ITD/TKD and FLT3 negative AML	MTD > 120 mg/daily CRc 91.3 % (FLT3 pos) and 56% (FLT3 neg)	Ongoing
02577406	Enasidenib vs. Conventional care	III	Advanced elderly IDH2 mutated AML	NA	Ongoing
02632708 [45]	Ivosidenib or Enasidenib with chemotherapy	I	Newly diagnosed, >18 years, IDH1-2 mutated AML	IVOSIDENIB arm: CR + CRi + CRp: 80% ENASIDENIB arm: CR + CRi + CRp: 72%	Active, not recruiting

NA—not available.

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
