# Peer review of "Acute Myeloid Leukemia Mutations: Therapeutic Implications"

_ijms, 2019, doi:10.3390/ijms20112721_

Round 1
Reviewer 1 Report
Authors did a great job. Authors complied the information very well and review is very informative. But it can be shortened possibly. Here are a few suggestions.
1) Title of the review is not connected to what the authors discussed in the review. They describe the molecular landscape of acute myeloid leukemia with therapeutic implications. But they give the title molecular pathway.
2) Add some new references.
3) In conclusion, add few more lines about future perspective.
Author Response
1) The title of the review has been changed to:
Acute Myeloid Leukemia mutations: therapeutic implications.
Therefore, the review is focused on the main mutations described in AML, and the recently approved drugs against these mutations
2) Some new references have been added
3) Conclusion chapter has been enriched with future perspectives
Reviewer 2 Report
This review article may merit the researcher in the relevant field. However, the paper is not well-organized. In the introduction section, the authors described that four drugs have been approved by FDA but did not mention gemtuzumab ozogamicin and CPX-351, without providing any reasons. To make the situation more confusing is that the authors did not discuss the strategy to use different types of drugs based on the molecular landscape of AML. The authors should discuss their own idea how to use these different types of molecular targeting drugs to treat AML.
The paper contains several unacceptably long paragraphs. For example, the paragraph from lines 75 to 131, should be divided into several paragraphs to enhance the readability. The authors should include the table(s) summarizing clinical trials to offer the overview of the present status of the drug development for AML to the readers.
Finally, the paper contains innumerable errors in syntax and wordings, and therefore, should be extensively edited by a professional editor who is proficient in writing medical English.
Author Response
The title of the review has been modified, and the authors decided to focus on mutations of AML and their therapeutic implications. Therefore the text has been shortened, and divided in smaller chapters. In the discussion part, the authors describe their own ideas on the introduction of these compounds in the clinical practice, as well as future perspectives.
Chapters have been shortened, and one table summarizing the most relevant trials (recently closed to enrollment or still ongoing) with drugs targeting mutations has been added
English has been reviewed by a professional editor
Round 2
Reviewer 2 Report
The review article is well-written and will merit the researchers in the relevant research fields.